# Factors associated with Catastrophic Healthcare Expenditure in communities of Lagos Nigeria: A Megacity experience

**Kikelomo Ololade Wright** [1,2]*, **Adeyinka Adeniran**[1,2], **Adedayo Aderibigbe**[2], **Olufunsho Akinyemi**[2], **Temiloluwa Fagbemi**[3], **Omoyeni Ayodeji**[3], **Biola Adepase**[4], **Emmanuella Zamba**[4], **Hussein Abdurrazzaq**[5], **Faith Oniyire**[5], **Olusegun Ogboye**[5], **Akin Abayomi**[5]

1 Department of Community Health and Primary Healthcare, Lagos State University College of Medicine, Lagos, Nigeria, 2 Department of Community Health and Primary Healthcare, Lagos State Teaching Hospital, Lagos, Nigeria, 3 Centre for Reproductive Health Research and Innovation, Lagos State University College of Medicine, Lagos, Nigeria, 4 Lagos State Ministry of Health, Lagos, Nigeria, 5 Lagos State Health Management Agency, Lagos, Nigeria

* kikelomo.wright@lasucom.edu.ng

**Data Availability Statement:** The datasets generated and analyzed during the current study are not publicly available because they contain potentially identifying or sensitive respondents'

## Abstract

### Background

Each year, millions of people in low—and middle-income countries such as Nigeria are forced into poverty and financial ruin due to out-of-pocket (OOP) healthcare expenses. Our study assessed the prevalence and determinants of Catastrophic Healthcare Expenditure (CHE) experienced by households in Lagos, Nigeria.

### Methods

A descriptive community-based cross-sectional survey was conducted on 2492 households in Lagos from December 2022 to March 2023 in 4 Local Government Areas (LGAs) using a multistage sampling technique. Data was collected using pre-tested semi-structured questionnaires, and analysis was performed using STATA 15.0 software. Univariate, bivariate, and binary logistic regression analyses were conducted with statistical significance set at p<0.05, and a 95% Confidence Interval was calculated for the adjusted odds ratio (OR).

### Results

The study revealed that 15% of households experience catastrophic health expenditure (CHE). Gender, marital status, educational level, occupation, personal income, health perception, household location, and health insurance enrollment were significantly associated with CHE. Additionally, gender, religion, income, household location, and self-rated health status were significant predictors of household CHE, with adjusted odds ratios of 4.42, 2.45, 1.00, 1.07, and 1.69, respectively.

information, but the Lagos State University Teaching Hospital—Health Research and Ethics Committee (LASUTH-HREC: Email: dcst@lasuth.org) can facilitate access upon reasonable request for researchers who meet the criteria for access to confidential data.

**Funding:** BILL AND MELINDA GATES FOUNDATION INV-016366 The funders had no role in study design, data collection and analysis, decision to publish, or preparation of the manuscript.

**Competing interests:** The authors have declared that no competing interests exist.

## Conclusion

CHE is more prevalent among people with lower socio-economic status, urban residents, and those lacking health insurance. It is crucial to implement targeted initiatives to raise awareness about the benefits of health insurance and simplify the enrollment process for vulnerable groups, thereby alleviating the financial strain of healthcare expenses.

## Introduction

Worldwide, universal health coverage (UHC) is crucial in achieving access to quality health care, a fundamental human right essential for positive health outcomes [1].

In developing nations such as Nigeria, practically all governments spend less than the prescribed 15% of their budget on health, which is inconsistent with the Abuja Declaration [2, 3].

Each year, 100 million people are forced into poverty due to out-of-pocket OOP expenses on health services, while 150 million people experience financial ruin [4]. Also, approximately 2 billion people are currently experiencing financial difficulties, of which 1 billion are facing high OOP medical expenses that can be catastrophic. OOP accounts for 32% of each country's health expenditure [1]. For instance, although healthcare in Nigeria is funded through various sources, the primary source of healthcare financing in the country is out-of-pocket spending, with over 70% of Nigerians depending on out-of-pocket healthcare services [5].

The World Health Organization recommends that health expenses should not exceed 10% of total household consumption or 25% of income to prevent various consequences such as catastrophic health expenditure (CHE) which generally refers to out-of-pocket (OOP) medical costs that exceed a household's financial capability level [6]. According to the World Health Organization (WHO) CHE is defined as out-of-pocket payments that exceed a certain threshold (40%) of a household's non-subsistence income [7–9].

In developed nations, several factors have been associated with catastrophic health expenses at the household level. These factors include insurance coverage, the age of household members, the presence of chronic illness in the household, the socioeconomic status and the use of healthcare facilities [9–11].

To protect its citizens from excessive healthcare expenditures, the Nigerian National Health Insurance Scheme (NHIS) was established in 1999 and introduced in 2005 to provide financial risk protection while reducing the heavy burden of OOP spending on individuals and households [12]. Under this scheme, state governments are also permitted to establish state-based health Insurance schemes. In 2007, Lagos State, often described as the commercial capital of Nigeria, launched Community Based Health Insurance schemes to provide social health protection coverage for the poor and underserved communities while the Lagos State Health Management Agency (LASHMA) was established to provide quality healthcare and reduce out-of-pocket spending. Subsequently, LASHMA launched the Lagos State Health Insurance Scheme in 2019. A recent study in Lagos revealed that only about 10.9% of respondents have ever been enrolled in health insurance. Among these, private health insurance schemes were the most popular, utilized by 58.5% of participants, while less than one-fifth (15.9%) opted for the Lagos State-owned scheme [13].

Several challenges, including poor health service delivery at public facilities, barriers to female education, fragmented public health insurance schemes, lack of political will, and low uptake of available health insurance, among others, are impeding the implementation of health insurance in Nigeria [1, 2, 14–17].

Studies have shown that the prevalence of catastrophic health expenditure in Nigeria varies between 2.5% and 44.0% at sub-national levels, particularly with the uneven distribution of resources across the country [10, 18].

Estimating healthcare financial spending at subnational levels is needed to implement the NHIS more efficiently and effectively, achieve wider coverage of universal healthcare, reduce CHE from OOP payments, and improve health outcomes.

As the most populous Nigerian city, the Lagos State government signed a law on establishing the Lagos State Health Management Agency (LASHMA) in 2015, to ensure that all state citizens have access to quality healthcare services without catastrophic financial outcomes [14, 19].

As part of efforts to guide decision-making, this study was conducted in Lagos to determine the prevalence of household CHE and its determinants among residents of Lagos, Nigeria.

## Materials and methods

Lagos State, situated in Nigeria's southwestern coastal area, has the smallest land mass in the country at the subnational level but constitutes the nation's most densely populated state, with an estimated 21 million residents and an annual growth rate between 4% and 8%. The state encompasses 3,577 square kilometers, resulting in a population density of approximately 16,067 individuals per square kilometer. It comprises 20 Local Government Areas (LGAs) divided into 4 rural and 16 urban LGAs with several peri-urban slums [4].

### Study design and population

This was a descriptive community-based cross-sectional study to assess CHE and its determinants among residents of Lagos State. The study participants include consenting individuals aged 18 years and above residing in Lagos at the time of the study.

The sample size was calculated using Fisher's formula for populations>10,000, using a standard normal deviation of 1.96, a proportion of respondents experiencing CHE in a previous study (60%), an 11 p-value of 0.05 for maximum variability, and an error margin of 0.05.

A minimum sample size of 369 was calculated for each Local Government Area (LGA) and adjusted to 554, assuming a design effect of 1.5, but increased to 1000 per LGA to further improve the power of this study. Therefore, the study used an estimated sample size 4000 for the selected 4 LGAs.

### Sampling method

A multi-stage sampling technique was used for the selection process. Firstly, a simple random technique was used to select four (3 urban and 1 rural) LGAs from a sampling frame of 20 (16 urban and 4 rural) LGAs in the state by balloting (Table 1). Secondly, one ward was randomly selected from each LGA. Using a sampling frame of all the streets in the selected wards, at least ten streets were chosen in the third stage. Thereafter, houses on each street were selected by

**Table 1. Sampling technique breakdown.**

| | Selection Criteria | Total Available | Selected | Selection Method |
|---|---|---|---|---|
| Stage 1: LGAs | Urban LGAs | 16 | 3 | Simple random sampling (balloting) |
| | Rural LGAs | 4 | 1 | Simple random sampling (balloting) |
| Stage 2: Wards | Wards within selected LGAs | Variable | 1 per LGA | Simple random sampling |
| Stage 3: Streets | Streets within selected wards | Variable | At least 10 per ward | Systematic random sampling |
| Stage 4: Houses | Houses on selected streets | Variable | Based on sampling interval of 2 | Systematic random sampling |
| Stage 5: Individuals | Consenting adults in households | Variable | 1 per household | Random selection (balloting) |

systematic random sampling based on the calculated sample interval. Adults aged 18 and older in each household were invited to participate in the study. If there were multiple households or consenting adults, one was selected by balloting.

## Study instrument

The research tool used in this study was a pretested questionnaire that included both open and closed-ended questions. The questionnaire was divided into three sections: the first section collected information on the socio-demographic and economic characteristics of the respondents, the second section focused on the respondents' general and healthcare expenditure, and the third section gathered information on the respondents' perceived health status and OOP payment for healthcare, including health insurance enrollment. The instrument was face-validated by all the investigators, and Cronbach's alpha reliability coefficient of 0.75 was computed.

## Data collection techniques

Data was collected between December 2022 and February 2023 by a team of 12 trained research assistants. The data collection tool was developed using REDCap at the Lagos State University College of Medicine (LASUCOM), allowing data entries through the downloaded REDCap app on electronic devices. To protect privacy, the collected data was anonymized, and access to data entry, editing, and administration was controlled, ensuring that only authorized personnel could view or alter sensitive information. Additionally, geo-coordinates were utilized to enhance the accuracy of the data collected, ensuring that the information was precise and reliable.

## Variables

The explanatory variables include age (in years), sex, marital status, the highest level of education, income level per month (Naira), household consumption expenditure, occupation which was classified into 5 (senior professionals such as academic professionals, senior admin executives; professionals such as teachers, nurses, etc.; skilled workers such as clerks, typists, etc; semi-skilled such as artisans, traders and lastly unskilled workers), location of household, whether rural or urban, enrollment in health insurance and current health status while the outcome variable was the presence or absence of CHE.

In this study, a respondent was considered to have incurred Catastrophic Health Expenditure (CHE) if their out-of-pocket (OOP) health expenditure, as a percentage of their annual income (excluding food expenditure), exceeded 40%. The 40% threshold was selected based on its widespread use in previous studies on catastrophic expenditure. The data collected includes healthcare expenses in Naira and the funding source for the treatment. All costs associated with healthcare visits to the hospital were documented to calculate the total health expenditure.

We estimated the annual household income by requesting details of monthly income from all income earners in the household and adding them together. Using appropriate multipliers, we estimated an average for those without a steady income and converted daily and weekly pay to monthly income. The annual income was calculated by multiplying the monthly income by 12 using the conversion rate of N780 to $1 (USD) at the time of the study (2023).

## Statistical analysis

Completed questionnaires from the REDCAP platform were cleaned and coded on Microsoft Excel 2018 and exported to STATA 15.0 software (StataCorp LLC Lakeway Drive, College

Station, Texas) for analysis. Data on sociodemographic information and CHE were presented using descriptive statistics. Bivariate analysis was done to determine the association between variables and CHE using significance tests.

CHE (catastrophic health expenditure) was defined as "not catastrophic = 0" and "catastrophic = 1" for bivariate analysis and logistic regression. The study looked at the relationship between explanatory variables such as age, gender, occupation, income, and the presence or absence of CHE. The significance level was set at 5% and logistic regression models that accounted for the survey design were used to identify the independent predictors of CHE. The logistic regression model included all significant variables in the bivariate analysis and presented as unadjusted odds ratios.

### Ethical considerations

Ethical approval was obtained from the Health Research Ethics Committee of the Lagos State University Teaching Hospital, Nigeria (LREC/06/10/1866). Each respondent provided written consent and was assured of the confidentiality of their information and their right to withdraw from the study at any point in time.

## Results

The study found that the mean age of the participants was 36.4 years (± 12.9), with 66.7% being below 40 years of age. More than half (56.5%) were married, and 68.6% had completed at least a secondary education while about one-fifth (20.6%) were non-skilled laborers, and 18.4% earned below the national minimum wage of N30,000. On average, each household had about 6 occupants, and a quarter of the respondents lived in rural areas of Lagos (Table 2).

Findings showed that the most common items respondents spent their income on were food, transportation and healthcare. The average monthly expenditure on these items in Nigerian Naira was ₦26,659 ($34.12), ₦12,240 ($15.69), and ₦6,274 ($8.04) for food, transportation, and healthcare, respectively (Table 3).

One-third of the respondents believed their health status was excellent, 41.0% said it was very good, while less than 1.0% rated their health status as poor. Hypertension accounted for 49.2% of all reported morbidity, while 36.0% of respondents were aware of health insurance, with 30% enrolled in any scheme. The average monthly healthcare spending was ₦6,221.38 ($7.78), funded primarily through cash savings (Table 4).

Only 15% of the participants experienced CHE in their households (Table 4, Fig 1).

Gender, marital status, educational level, occupation, personal income, perception of current health status, location of a household, and enrolment in health insurance schemes were significantly associated with CHE. Females (18.7%) experienced more CHE compared to males (11.4%), and married individuals (17.5%) experienced more CHE compared to singles (11.2%) (P<0.05). Semi-skilled respondents (19.4%) experienced more CHE than skilled respondents (16.3%) or senior professionals (10.5%) while respondents with personal income less than ₦30,000 ($38.5) experienced higher CHE (36.5%) compared to those who earned higher. Additionally, the household's location and health insurance enrollment were significantly associated with CHE. A higher proportion of respondents in the urban areas (17.6%) were more likely to experience CHE compared to those in the rural areas (7.1%), (P<0.05). Also, a higher proportion of households not enrolled in any health insurance schemes (14.1%) were more likely to experience CHE compared to those who were enrolled (7.4%) (P<0.05). Respondents' perceptions of current health status were not significantly associated with CHE (P>0.05) (Table 5).

**Table 2. Sociodemographic data of respondents.**

| Variables | Frequency (N = 2492) | Percentage |
|---|---|---|
| **Age** | | |
| <30 | 859 | 34.5% |
| 30–40 | 802 | 32.2% |
| 41–50 | 477 | 19.1% |
| 51–60 | 222 | 8.9% |
| >60 | 129 | 5.2% |
| Non-response | 3 | 0.1 |
| Mean ± SD | 36.35 ± 12.85 | |
| **Gender** | | |
| Male | 1260 | 50.6% |
| Female | 1230 | 49.3% |
| Non-response | 2 | 0.1% |
| **Marital Status** | | |
| Single | 926 | 37.2% |
| Married | 1406 | 56.4% |
| Widow/Widower | 121 | 4.9% |
| Others | 36 | 1.4% |
| Non-response | 2 | 0.1% |
| **Education Level** | | |
| Primary | 233 | 9.3% |
| Secondary | 1475 | 59.2% |
| Tertiary | 635 | 25.5% |
| Postgraduate | 77 | 3.1% |
| No Formal Education | 70 | 2.8% |
| Non-response | 2 | 0.1% |
| **Occupation** | | |
| Senior professionals, | 77 | 3.1% |
| Professionals | 311 | 12.5% |
| Skilled workers | 731 | 29.3% |
| Semi-skilled | 413 | 16.3% |
| Unskilled workers | 514 | 20.6% |
| Non-response | 446 | 17.9% |
| **Monthly Income (₦)** | | |
| <30,000 | 458 | 18.4% |
| 30,000–50,000 | 852 | 34.3% |
| 50,001–100,000 | 849 | 34.2% |
| >100,000 | 324 | 13.0% |
| Mean ± SD | 72750.11 ± 109518.35 | |
| **Location of household** | | |
| Rural | 623 | 25.0% |
| Urban | 1866 | 74.9% |
| Non-response | 2 | 0.1% |
| **Number of people in the household** | | |
| <4 | 1126 | 45.2% |
| 4–6 | 1267 | 50.9% |
| >6 | 96 | 3.9% |
| Mean ± SD | 6.39 ± 121.19 | |

**Table 3. Consumption expenditure of households.**

| Variables | Mean ± SD | Median (Min-Max) | 25% -75% |
|---|---|---|---|
| Food | 26659.97 ± 21408.93 | 25000.00(0.00–300000.00) | 15000.00–30000.00 |
| Education | 2.31 ± 0.79 | 2.00(1.00–5.00) | 2.00–3.00 |
| Healthcare | 6274.80 ± 13523.88 | 4000.00(0.00–450000.00) | 2000.00–6000.00 |
| Recreation | 3531.60 ± 7815.25 | 0.00(0.00–100000.00) | 0.00–4000.00 |
| Transportation | 12240.38 ± 16254.73 | 6000.00(0.00–300000.00) | 2800.00–15000.00 |
| Communication | 5721.99 ± 8474.54 | 4000.00(0.00–250000.00) | 2000.00–6000.00 |
| Cloths | 5322.25 ± 9327.68 | 3000.00(0.00–150000.00) | 0.00–6000.00 |
| Smoking | 549.85 ± 5312.05 | 0.00(0.00–250000.00) | 0.00–0.00 |
| Alcohol | 2997.97 ± 7822.18 | 0.00(0.00–100000.00) | 0.00–2000.00 |

The study found that gender, religion, income, household location, and respondents' rating of health morbidity were significant predictors associated with household CHE with an adjusted odds ratio (AOR) of 4.42, 2.45, 1.00, 1.07, and 1.69, respectively, at p<0.001 (Table 6).

## Discussion

The consequences of CHE in Lagos can lead to not only poor health outcomes but also a decline in the socio-economic status of families and households that are on the verge of poverty. In this study, the average monthly expenditure on food items was ₦26,659 ($34.12) and this was slightly similar to a previous conducted in Lagos State, which reported an average food expenditure of ₦29,282 ($37.54) [11].

At a 40% threshold of non-food expenditure, a CHE prevalence of 15% was found in our study, which is slightly higher than figures reported in studies conducted in other low-socio-economic countries such as Burkina Faso (10.8%) and Kenya (9.8%) [20, 21]. Another Nigerian study also reported catastrophic health spending of 13.7% [22]. Our study shows a higher CHE than another Kenyan study, which indicated a prevalence of 1.55% at a 30% threshold using the WHO model and potentially a lower prevalence of under 1.52% at a 40% threshold. This difference may be explained by the monthly OOP payments of ₦6,221.38 in our study compared to the average OOP health spending of 337.7 Kenyan Shillings (approximately ₦4,269.99) in the Kenyan study [23].

Recent reports indicate that there has been an increase in healthcare expenditure (CHE) patterns in sub-Saharan Africa. This trend is mainly due to factors such as the higher cost of living, rising healthcare costs, a growing number of chronic ailments coupled with prevalent infectious diseases, and the recent COVID-19 pandemic, which may have further amplified the vulnerability of health systems [24].

Our study revealed that there is a difference in the prevalence of CHE between urban and rural households in Lagos. The study found that households in urban areas had a higher prevalence of CHE (17.6%) than those in rural areas (7.1%). This finding differs from the findings of two other studies conducted in Ekiti and Ibadan, also located in southwest Nigeria, which reported a higher prevalence of CHE in rural areas than in urban settings [25–27]. Urban Lagos is a densely populated metropolis that serves as the economic hub of the country. The high population density has led to a correspondingly high cost of living, making the healthcare system overburdened. Consequently, the cost of healthcare services in urban areas is likely to be higher than their rural counterparts.

The study findings indicate that there is an association between CHE and various sociodemographic and socioeconomic factors such as age, gender, educational level, occupation, and

**Table 4. Respondents' health status and their perception.**

| VARIABLE | Freq. (n = 2492) | Percentage |
|---|---|---|
| **Respondents' rating of current health status** | | |
| Excellent | 804 | 32.3% |
| Very good | 1022 | 41.0% |
| Good | 571 | 22.9% |
| Fair | 86 | 3.5% |
| Poor | 6 | 0.2% |
| Non-response | 3 | 0.1% |
| **Any morbidity (known health condition)** | | |
| Yes | 313 | 12.6% |
| No | 2093 | 84.0% |
| Don't know | 83 | 3.3% |
| Non-response | 3 | 0.1% |
| **Morbidity type** | | |
| Diabetes | 86 | 27.5% |
| Hypertension | 154 | 49.2% |
| Others* | 73 | 23.3% |
| **Recent illness in past 4 weeks (Acute)** | | |
| Yes | 332 | 13.3% |
| No | 2154 | 86.4% |
| Non-response | 6 | 0.2% |
| **Respondent/ immediate family received care from health provider in the last 4 weeks.** | | |
| Yes | 577 | 23.2% |
| No | 1910 | 76.6% |
| Non-response | 5 | 0.2% |
| **Illness duration (days)** | | |
| Mean ± SD | 3.52 ± 3.34 | |
| **Amount spent on treatment and services (Naira)** | | |
| Mean ± SD | 10282.00 ± 23035.72 | |
| **Estimated monthly household spending on healthcare (Naira)** | | |
| Mean ± SD | 6221.38 ± 9456.61 | |
| **Source(s) of money to meet costs of treatment (coping with cost)** | | |
| Cash savings | 1917 | 77.1% |
| Borrowing | 145 | 5.8% |
| Gifts | 139 | 5.6% |
| Credit | 30 | 1.2% |
| Sell assets | 4 | 0.2% |
| Health Insurance | 187 | 7.5% |
| Others | 65 | 2.6% |
| **Hospitalization (at least overnight stay in the last 6 months)** | | |
| Yes | 307 | 12.3% |
| No | 2179 | 87.4% |
| Non-response | 6 | 0.2% |
| **Ever enrolled in Health Insurance Scheme** | | |
| Yes | 270 | 10.8% |
| No | 2222 | 89.2% |
| **Household spending on healthcare in last 12 months–(Naira)** | | |

*(Continued)*

**Table 4.** (Continued)

| VARIABLE | Freq. (n = 2492) | Percentage |
|---|---|---|
| Mean ± SD | 29638.06 ± 53471.70 | |
| Median (Min-Max) | 15000(0.00–100000) | |
| 25% -75% | 6000.00–30000.00 | |
| Experienced CHE | 373 (15%) | |

*Others include renal, musculoskeletal, respiratory diseases, etc.

income. It was observed that individuals belonging to households with lower levels of education and income experienced higher levels of CHE. Moreover, people with better socioeconomic status may have better healthcare-seeking behavior and can afford healthcare services as observed from findings in previous studies [22, 28, 29]. Approximately 35% of the participants in the study were under the age of 30 and had lower CHE. This age group is less likely to have chronic health conditions that require frequent healthcare visits and spending, which could be the reason for the lower CHE in this group.

The socioeconomic inequalities related to CHE can be reduced through financial risk protection of health insurance schemes. However, less than 10% of respondents who experienced CHE were enrolled in any health insurance scheme that could have offered protection. This study revealed that health insurance status has a statistically significant association with catastrophic health expenditure among respondents, contrasting with the findings from a study in Iran [11, 30].This observation confirmed that health insurance protects against CHE.

Respondents who rated their current health status as poor experienced catastrophic health expenditure more than those who rated their health as fair or excellent, though there is no statistically significant association. Those with poor health status may have higher healthcare costs, similar to a study done in Lagos among persons living with HIV (PLHIV) [11].

In consonance with other studies, our findings showed that factors such as gender, religion, income, location of household and the perception of health morbidity predict the presence of CHE [11, 16, 26, 31].

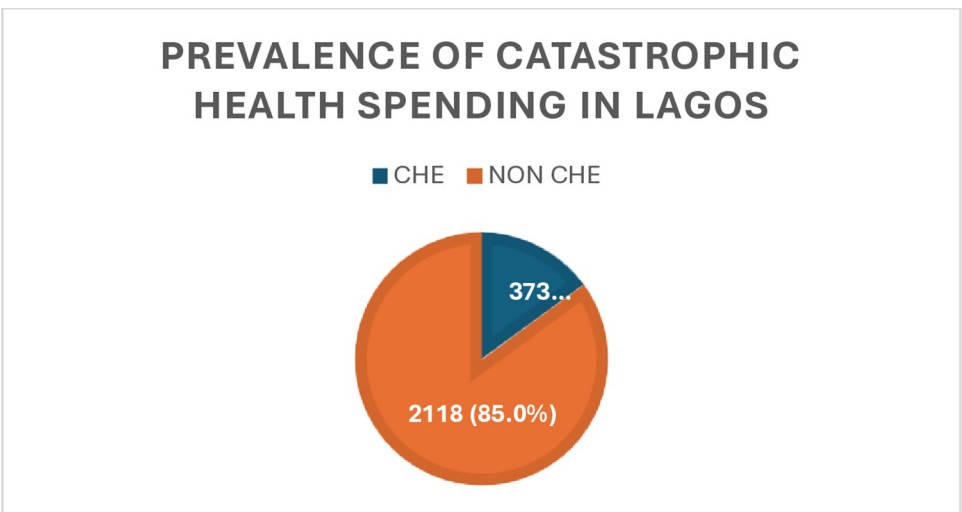

**Fig 1. The prevalence of catastrophic health spending within households in Lagos was 15% (373).**

**Table 5. Relationship between CHE and sociodemographic characteristics of respondents.**

| Sociodemographic Variable | Non-CHE | CHE | Statistic | P-value |
|---|---|---|---|---|
| **AGE** | | | | |
| <30 | 749(87.2) | 110(12.8) | $X^2 = 9.2$ | 0.056 |
| 30–40 | 678(84.6) | 123(15.4) | | |
| 41–50 | 404(84.7) | 73(15.3) | | |
| 51–60 | 176(79.3) | 46(20.7) | | |
| >60 | 108(83.7) | 21(16.3) | | |
| **Gender** | | | | |
| Male | 1116(88.6) | 143(11.4) | $X^2 = 26.3$ | <0.001** |
| Female | 1000(81.3) | 230(18.7) | | |
| **Marital Status** | | | | |
| Single | 822(88.8) | 104(11.2) | $X^2 = 17.7$ | 0.001* |
| Married | 1159(82.5) | 246(17.5) | | |
| Widow/Widower | 102(84.3) | 19(15.7) | | |
| Others | 32(88.9) | 4(11.1) | | |
| **Education Level** | | | | |
| Primary | 191(82.0) | 42(18.0) | $X^2 = 17.6$ | 0.002* |
| Secondary | 1231(83.5) | 244(16.5) | | |
| Tertiary | 572(90.1) | 63(9.9) | | |
| Postgraduate | 63(82.9) | 13(17.1) | | |
| No Formal Education | 59(84.3) | 11(15.7) | | |
| **Occupation** | | | | |
| Senior professionals | 68(89.5) | 8(10.5) | $X^2 = 16.9$ | 0.002* |
| Professionals | 283(91.0) | 28(9.0) | | |
| Skilled workers | 612(83.7) | 119(16.3) | | |
| Semi-skilled workers | 333(80.6) | 80(19.4) | | |
| Unskilled workers | 438(85.2) | 76(14.8) | | |
| **Monthly Income (₦)** | | | | |
| <30,000 | 290(63.5) | 167(36.5) | $X^2 = 239.2$ | <0.001* |
| 30,000–50,000 | 726(85.2) | 126(14.8) | | |
| 50,001–100,000 | 780(91.9) | 69(8.1) | | |
| >100,000 | 317(97.5) | 8(2.5) | | |
| **Location of household** | | | | |
| Rural | 579(92.9) | 44 (7.1) | $X^2 = 41.0$ | <0.001** |
| Urban | 1536(82.4) | 329 (17.6) | | |
| **Rating current health status** | | | | |
| Excellent | 690(85.9) | 113(14.1) | $X^2 = 7.0$ | 0.135 |
| Very good | 873(85.4) | 149(14.6) | | |
| Good | 482(84.4) | 89(15.6) | | |
| Fair | 66(76.7) | 20(23.3) | | |
| Poor | 4(66.7) | 2(33.3) | | |
| **Known health condition** | | | | |
| Yes | 268(85.6) | 45(14.4) | $X^2 = 0.3$ | 0.847 |
| No | 1775(84.8) | 317(15.2) | | |
| Don't know | 72(86.7) | 11(13.3) | | |
| **Ever enrolled in health insurance scheme** | | | | |
| Yes | 250(92.6) | 20(7.4) | $X^2 = 8.0$ | 0.003** |

*(Continued)*

**Table 5.** (Continued)

| Sociodemographic Variable | Non-CHE | CHE | Statistic | P-value |
|---|---|---|---|---|
| No | 542(85.9) | 89(14.1) | | |

"*" means significant P-value while

"**" means Fisher's exact value

## Conclusion

Our study has shown that CHE is more common among individuals living in urban areas, those with low socio-economic status, and those without health insurance. To mitigate the financial burden of healthcare costs, it is essential to raise awareness of the benefits of health insurance and actively encourage enrolment among vulnerable groups, particularly those in lower economic strata, in both formal and informal sectors. This collective effort can effectively reduce out-of-pocket expenses and prevent the negative impact of CHE on individuals and families.

## Strengths and limitations of the study

The community-based nature and the large sample size are considered to be the major strengths that enhance the quality and robustness of our study findings.

Recall bias was minimized by targeting questions about illness and care 4 weeks before the interview. Social desirability bias was also a limitation, which was reduced by making respondents self-report on questions on household income.

## Implication for policy and future research

Increasing awareness and promoting enrollment in health insurance schemes, especially among vulnerable groups like those with low socioeconomic status and those living in urban areas can help reduce CHE from OOP payments. Also, addressing the socioeconomic inequalities that contribute to higher CHE, such as improving access to education and employment opportunities, which can increase household income levels; strengthening the implementation

**Table 6.  CHE predictors among respondents.**

| Variables | Adjusted OR | 95% CI | P-value |
|---|---|---|---|
| Age | 1.01 | 0.99–1.04 | 0.204 |
| Gender | 4.42 | 2.12–9.21 | <0.001* |
| Religion | 2.45 | 1.56–3.86 | <0.001* |
| Employment | 0.67 | 0.39–1.13 | 0.129 |
| Education | 1.08 | 0.79–1.47 | 0.644 |
| Income Grp, | 1.00 | 1.00–1.00 | <0.001* |
| Occupation | 1.17 | 0.91–1.51 | 0.221 |
| Location of household | 1.07 | 1.01–1.14 | 0.028* |
| Rate of health morbidity | 1.69 | 1.25–2.29 | 0.001* |
| Head of household | 3.01 | 0.22–42.02 | 0.413 |
| Education of head | 0.78 | 0.60–1.03 | 0.077 |
| Constant | 0.03 | 0.002–0.27 | 0.002* |

* 5% significance level

and coverage of the National Health Insurance Scheme (NHIS) and the Lagos State Health Management Agency (LASHMA) health insurance programs to provide better financial risk protection to households; exploring ways to subsidize or provide financial assistance for healthcare costs to households below a certain income threshold to prevent catastrophic expenditures and impoverishment due to medical expenses are some of the policy implications of this study. Further research at subnational or national levels will be valuable in better understanding the determinants and patterns of CHE and evaluating targeted policy interventions.

## Acknowledgments

The authors thank the Lagos State Ministry of Health and LASHMA for their support in completing this survey, and the participants for their willingness to participate.

## Author Contributions

**Conceptualization:** Kikelomo Ololade Wright, Emmanuella Zamba, Hussein Abdurrazzaq, Olusegun Ogboye, Akin Abayomi.

**Data curation:** Adeyinka Adeniran, Adedayo Aderibigbe, Olufunsho Akinyemi, Temiloluwa Fagbemi, Omoyeni Ayodeji.

**Formal analysis:** Kikelomo Ololade Wright, Adeyinka Adeniran, Adedayo Aderibigbe, Olufunsho Akinyemi, Temiloluwa Fagbemi, Omoyeni Ayodeji.

**Funding acquisition:** Biola Adepase, Emmanuella Zamba, Hussein Abdurrazzaq, Faith Oniyire, Olusegun Ogboye, Akin Abayomi.

**Investigation:** Biola Adepase.

**Methodology:** Kikelomo Ololade Wright, Adeyinka Adeniran, Temiloluwa Fagbemi, Faith Oniyire, Akin Abayomi.

**Project administration:** Kikelomo Ololade Wright, Adedayo Aderibigbe, Temiloluwa Fagbemi, Omoyeni Ayodeji, Biola Adepase, Hussein Abdurrazzaq, Faith Oniyire.

**Resources:** Biola Adepase, Emmanuella Zamba, Hussein Abdurrazzaq, Faith Oniyire, Olusegun Ogboye, Akin Abayomi.

**Software:** Kikelomo Ololade Wright, Adeyinka Adeniran.

**Supervision:** Kikelomo Ololade Wright, Adedayo Aderibigbe, Omoyeni Ayodeji, Hussein Abdurrazzaq.

**Validation:** Temiloluwa Fagbemi.

**Writing – original draft:** Kikelomo Ololade Wright, Adeyinka Adeniran, Olufunsho Akinyemi.

**Writing – review & editing:** Kikelomo Ololade Wright, Adeyinka Adeniran, Adedayo Aderibigbe, Olufunsho Akinyemi, Temiloluwa Fagbemi, Omoyeni Ayodeji, Biola Adepase, Emmanuella Zamba, Hussein Abdurrazzaq, Faith Oniyire, Olusegun Ogboye, Akin Abayomi.

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
