## [Decision Letter · Decision Letter 0]

4 Oct 2024

PONE-D-24-25551Factors Associated with Catastrophic Healthcare Expenditure in Communities of Lagos Nigeria: A Megacity ExperiencePLOS ONE

Dear Dr. Wright,

Thank you for submitting your manuscript to PLOS ONE. After careful consideration, we feel that it has merit but does not fully meet PLOS ONE’s publication criteria as it currently stands. Therefore, we invite you to submit a revised version of the manuscript that addresses the points raised during the review process.

We look forward to receiving your revised manuscript.

Kind regards,

Ugochukwu Anthony Eze

Academic Editor

PLOS ONE

Journal Requirements:

2. Thank you for stating the following financial disclosure: BILL AND MELINDA GATES FOUNDATION

INV-016366 

Reviewers' comments:

Reviewer's Responses to Questions

**Comments to the Author**

1. Is the manuscript technically sound, and do the data support the conclusions?

Reviewer #1: Yes

Reviewer #2: Yes

2. Has the statistical analysis been performed appropriately and rigorously? 

Reviewer #1: Yes

Reviewer #2: Yes

3. Have the authors made all data underlying the findings in their manuscript fully available?

Reviewer #1: Yes

Reviewer #2: Yes

4. Is the manuscript presented in an intelligible fashion and written in standard English?

Reviewer #1: Yes

Reviewer #2: Yes

5. Review Comments to the Author

Reviewer #1: Title – Factors Associated with Catastrophic Healthcare Expenditure in Communities of Lagos

Nigeria: A Megacity Experience

- Appropriate

Abstract – Well presented.

Introduction – Statement of problem, magnitude of problem, rationale for the study and study objectives well presented.

Methods and analysis – Clearly presented. Study area description, study population, sample size calculation, sampling technique, data collection, variables and data analysis are well presented.

Study setting, target population, sampling techniques and data management are well described

•There are too many 1 - sentence paragraphs. Avoid 1 - sentence paragraphs

Result

Table 4 – replace < 0.000 with < 0.001

Discussion – Findings well discussed.

Line 231 – The CHE cannot be regarded as similar (Over 50% higher). It is better referred to as "higher"

Line 235 – Compare the OOP in Kenya with Nigeria before this statement can be affirmed.

Conclusion and Recommendations are appropriate

References - Adequate

Reviewer #2: 1. The manuscript is technically sound.

2.. The choices of the statistical tools were appropriate for the Questions

3. All data was made available

4.The English language was presented correctly.

5. The article should be accepted.

6. PLOS authors have the option to publish the peer review history of their article (what does this mean?). If published, this will include your full peer review and any attached files.

Reviewer #1: **Yes: **Prof Tanimola Makanjuola Akande

Reviewer #2: No

---

## [Author Response · Author response to Decision Letter 0]

9 Oct 2024

Point-by-point responses to reviewers' comments.

Reviewer #1: 

Title – Factors Associated with Catastrophic Healthcare Expenditure in Communities of Lagos

Nigeria: A Megacity Experience

- Appropriate

Abstract – Well presented.

Introduction – Statement of problem, magnitude of problem, rationale for the study and study objectives well presented.

Methods and analysis – Clearly presented. Study area description, study population, sample size calculation, sampling technique, data collection, variables and data analysis are well presented.

Study setting, target population, sampling techniques and data management are well described. Thank you

Response To Comments

1. •There are too many 1 - sentence paragraphs. Avoid 1 - sentence paragraphs

Result:

Table 4 – replace < 0.000 with < 0.001

This has been done

Discussion – Findings well discussed.

Thank you

2. Line 231 – The CHE cannot be regarded as similar (Over 50% higher). It is better referred to as "higher." 

This has been corrected to read ‘’which is slightly higher than figures’’ in line 243

3. Line 235 – Compare the OOP in Kenya with Nigeria before this statement can be affirmed. 

This has been corrected to read, “Our study shows a higher CHE than another Kenyan study, which indicated a prevalence of 1.55% at a 30% threshold using the WHO model and potentially a lower prevalence of under 1.52% at a 40% threshold. This difference may be explained by the monthly OOP payments of ₦6,221.38 in our study compared to the average OOP health spending of 337.7 Kenyan Shillings (approximately ₦4,269.99) in the Kenyan study. 23” Lines 245-250

Conclusion and Recommendations are appropriate. Thank you

References – Adequate. Thank you

4. Data Availability statement

The datasets generated and analyzed during the current study are not publicly available because they contain potentially identifying or sensitive respondents’ information, but the Lagos State University Teaching Hospital—Health Research and Ethics Committee (LASUTH-HREC) can facilitate access upon reasonable request. (Email: dcst@lasuth.org) Lines 325-328

5. Role the funders took in the study.

Reviewer #2: Thank you

1. The manuscript is technically sound.

2.. The choices of the statistical tools were appropriate for the Questions

3. All data was made available

4. The English language was presented correctly.

5. The article should be accepted. Thank you

---

## [Decision Letter · Decision Letter 1]

8 Dec 2024

PONE-D-24-25551R1Factors Associated with Catastrophic Healthcare Expenditure in Communities of Lagos Nigeria: A Megacity ExperiencePLOS ONE

Dear Dr. Wright,

Thank you for submitting your manuscript to PLOS ONE. After careful consideration, we feel that it has merit but does not fully meet PLOS ONE’s publication criteria as it currently stands. Therefore, we invite you to submit a revised version of the manuscript that addresses the points raised during the review process.

**ACADEMIC EDITOR:** Thank you for submitting the revised paper. There are few items to address as stated below. Kindly review and address all the comments. **Introduction:**

Will be appropriate if the authors can provide insight about the status of health insurance in the study state including coverage, distribution and progress made by LASHMA.

**Methods:**

Sampling methods:

Will be good to provide a table illustrating the numbers at each stage of the selection LGAs, wards, streets e.t.c.

Lines 153 – 154: please indicate the year of the conversion rate. Also indicate that it is US Dollars.

Data collection:

Please indicate the periods when the data was collected, who collected the data and how the data was transmitted into REDCAP. Also highlight how data confidentiality and security was enhanced. Remember, the data was sensitive and was not shared in this submission which implies that it requires protection and security.

Results:

Figure 1 may not be necessary since it does not add any additional value. The findings presented in the figure have been outlined in the results. Also there are only 2 variables. However, if the authors decided to retain the figure, then clearly present the caption and define the abbreviations.

Acknowledgement:

Please include this section at the end of the paper.

Kind regards,

Ibrahim Jahun, MD, MSC, PhD

Academic Editor

PLOS ONE

Journal Requirements:

Reviewers' comments:

Reviewer's Responses to Questions

**Comments to the Author**

1. If the authors have adequately addressed your comments raised in a previous round of review and you feel that this manuscript is now acceptable for publication, you may indicate that here to bypass the “Comments to the Author” section, enter your conflict of interest statement in the “Confidential to Editor” section, and submit your "Accept" recommendation.

Reviewer #1: All comments have been addressed

Reviewer #2: All comments have been addressed

2. Is the manuscript technically sound, and do the data support the conclusions?

Reviewer #1: Yes

Reviewer #2: Yes

3. Has the statistical analysis been performed appropriately and rigorously? 

Reviewer #1: Yes

Reviewer #2: Yes

4. Have the authors made all data underlying the findings in their manuscript fully available?

Reviewer #1: Yes

Reviewer #2: Yes

5. Is the manuscript presented in an intelligible fashion and written in standard English?

Reviewer #1: Yes

Reviewer #2: Yes

6. Review Comments to the Author

Reviewer #1: All the corrections have been effected by the authors. No additional comments to be addressed by the authors

Reviewer #2: The Manuscript is intelligently written and technically sound.

the Statistical analysis had answered the research questions.

7. PLOS authors have the option to publish the peer review history of their article (what does this mean?). If published, this will include your full peer review and any attached files.

Reviewer #1: **Yes: **Prof. Tanimola Makanjuola Akande

Reviewer #2: **Yes: **Bello Shahir Umar

---

## [Author Response · Author response to Decision Letter 1]

14 Dec 2024

Point-by-point responses to reviewers' comments.

Introduction: 

Will be appropriate if the authors can provide insight about the status of health insurance in the study state including coverage, distribution and progress made by LASHMA. 

Lines 78-87

Under this scheme, state governments are also permitted to establish state-based health Insurance schemes. In 2007, Lagos State, often described as the commercial capital of Nigeria launched Community Based Health Insurance schemes to provide social health protection coverage for the poor and underserved communities while the Lagos State Health Management Agency (LASHMA) was established to provide quality healthcare and reduce out-of-pocket spending. Subsequently, LASHMA launched the Lagos State Health Insurance Scheme in 2019. A recent study conducted in Lagos revealed that only about 10.9% of respondents have ever been enrolled in any type of health insurance. Among these, private health insurance schemes were the most popular, utilized by 58.5% of participants, while less than one-fifth (15.9%) opted for the Lagos State-owned scheme. 13 

Methods: 

Sampling methods: 

• 

Will be good to provide a table illustrating the numbers at each stage of the selection LGAs, wards, streets etc. 

• 

 Lines 131-132

Stage Selection Criteria Total Available Selected Selection Method 

Stage 1: LGAs Urban LGAs 16 3 Simple random sampling (balloting) 

 Rural LGAs 4 1 Simple random sampling (balloting) 

Stage 2: Wards Wards within selected LGAs Variable 1 per LGA Simple random sampling 

Stage 3: Streets Streets within selected wards Variable At least 10 per ward Systematic random sampling 

Stage 4: Houses Houses on selected streets Variable Based on sampling interval of 2 Systematic random sampling 

Stage 5: Individuals Consenting adults in households Variable 1 per household Random selection (balloting) 

Lines 153 – 154: please indicate the year of the conversion rate. Also indicate that it is US Dollars. 

DONE 

Data collection: 

• 

Please indicate the periods when the data was collected, who collected the data and how the data was transmitted into REDCAP. Also highlight how data confidentiality and security was enhanced. Remember, the data was sensitive and was not shared in this submission which implies that it requires protection and security. 

• 

DONE (Lines 143-148)

Results: 

• 

Figure 1 may not be necessary since it does not add any additional value. The findings presented in the figure have been outlined in the results. Also there are only 2 variables. However, if the authors decided to retain the figure, then clearly present the caption and define the abbreviations. 

• 

FIGURE 1 HAS BEEN REMOVED 

Acknowledgement: 

Acknowledgements (Lines 327-329)

Please include this section at the end of the paper. 

Additional reference: 

13. Adeniran A, Wright K, Aderibigbe A, Akinyemi O, Fagbemi T, Ayodeji O, Adepase A, Zamba E, Abdur-Razzaq H, Oniyire F, Ogboye O, Abayomi A. Determinants of health insurance adoption among residents of Lagos, Nigeria: A cross-sectional survey. Open Health. 2024;5(1): 20230043. https://doi.org/10.1515/ohe-2023-0043

---

## [Editor Report · Decision Letter 2]

17 Dec 2024

Factors Associated with Catastrophic Healthcare Expenditure in Communities of Lagos Nigeria: A Megacity Experience

PONE-D-24-25551R2

Dear Dr. Wright,

We’re pleased to inform you that your manuscript has been judged scientifically suitable for publication and will be formally accepted for publication once it meets all outstanding technical requirements.

Kind regards,

Ibrahim Jahun, MD, MSc, PhD

Academic Editor

PLOS ONE
---

## [Editor Report · Acceptance letter]

2 Jan 2025

PONE-D-24-25551R2 

PLOS ONE

Dear Dr. Wright, 

I'm pleased to inform you that your manuscript has been deemed suitable for publication in PLOS ONE. Congratulations! Your manuscript is now being handed over to our production team.

Kind regards, 

on behalf of

Dr. Ibrahim Jahun 

Academic Editor

PLOS ONE